# Impact of Pneumococcal Vaccination in the Nasopharyngeal Carriage of Streptococcus pneumoniae in Healthy Children of the Murcia Region in Spain

**DOI:** 10.3390/vaccines9010014

**Published:** 2020-12-28

**Authors:** Santiago Alfayate Miguélez, Genoveva Yague Guirao, Ana I. Menasalvas Ruíz, Manuel Sanchez-Solís, Mirian Domenech Lucas, Fernando González Camacho, M. Mar Ortíz Romero, Pilar Espejo García, Carmen Guerrero Gómez, Antonio Iofrío de Arce, Laura Moreno Parrado, Rosa M. Sánchez Andrada, Eva Cascales Alcolea, Sebastián Lorente García, Pedro Paredes Reyes, Ángela Casquet Barceló, M. Luisa López Yepes, Juan José Vigueras Abellán, Gonzalo Sanz Mateo

**Affiliations:** 1Pediatrics Infectious Disease and Virgen de la Arrixaca Hospital, 30120 Murcia, Spain; amenasalvas@hotmail.com (A.I.M.R.); msolis@um.es (M.S.-S.); 2Microbiology Department, Virgen de la Arrixaca Hospital, 30120 Murcia, Spain; pedror.paredes@carm.es; 3Pneumoccocal Reference Laboratory CNM-ISCIII, 28220 Madrid, Spain; mdomenech@isciii.es (M.D.L.); fgonzalezc@isciii.es (F.G.C.); 4Microbiology Department S, Lucía Hospital, 30202 Cartagena, Spain; mmar.ortiz2@carm.es; 5Barrio del Peral Health care Center, 30300 Cartagena, Spain; mpespejo@hotmail.com; 6Microbiology Department Morales Meseguer Hospital, 30008 Murcia, Spain; carsmile@hotmail.com; 7El Ranero Health care Center, 30009 Murcia, Spain; aiofrio@gmail.com; 8Parrado Moreno L Los Arcos-Mar Menor Hospital, 30739 San Javier, Spain; lauramorenoparrado@hotmail.com; 9San Javier Health care Center, 30730 San Javier, Spain; rosandrada@yahoo.es; 10Microbiology Department Rafael Méndez Hospital, 30739 San Javier, Spain; eva.cascales@carm.es; 11Totana Health care Center, 30850 Murcia, Sapin; sebastian.lorente@carm.es; 12Ronda Sur. Health care Center, 30010 Murcia, Spain; angelacasquet_barcelo@hotmail.com; 13Microbiology Department V del Castillo Hospital, 30510 Yecla, Spain; mluisa.lopez3@carm.es; 14Pediatrician. Centro de Salud Mariano Yago, 30510 Yecla, Spain; jjva70@hotmail.com; 15Santa María de Gracia Health care Center, 30009 Murcia, Spain; gzmateo@hotmail.com

**Keywords:** Streptococcus pneumoniae nasopharyngeal carriage, conjugate vaccines, antimicrobial susceptibility

## Abstract

Background: An epidemiological study of Streptococcus pneumoniae nasopharyngeal carriage in healthy children was carried out five years after the introduction of the 13-valent pneumococcal conjugate vaccine (PCV13). Objectives: Study the impact of pediatric vaccination with PCV13, and other associated epidemiological factors on the status of nasopharyngeal carriage, the circulating pneumococcal serotypes, and the antibiotic susceptibility to more frequently used antibiotics. Methods: A multi-center study was carried out in Primary Health Care, which included 1821 healthy children aged 1 to 4 years old. All isolates were sent to the Spanish Pneumococcal Reference Laboratory for serotyping and antimicrobial susceptibility testing. Results: At least one dose of PCV13 had been received by 71.9% of children and carriage pneumococcal prevalence was 19.7%. The proportion of PCV13 serotypes was low (14.4%), with an observed predominance of non-vaccine serotypes, 23B, 11A, 10A, 35B/F, and 23A were the five most frequent. A high rate of resistance to penicillin, erythromycin, and trimethoprim sulfamethoxazole was found. Conclusions: A low proportion of PCV13 serotypes were detected, confirming the impact of pediatric vaccination for reducing the serotypes vaccine carriage. High resistance rates to clinically important antibiotics were observed.

## 1. Introduction

Streptococcus pneumoniae is an important pathogen that causes invasive and non-invasive diseases such as otitis, sepsis, pneumonia, and meningitis. Invasive pneumococcal disease (IPD) remains a global health problem, even in developed countries. S. pneumoniae is a human pathogen than colonizes the nasopharynx, and this asymptomatic nasopharyngeal carriage (NPC) is considered a prerequisite for the development of diseases [1,2,3]. NPC is high in children (about 30%), especially in those aged less than five years [1,2,3], and has a key role in transmission to the rest of the population [1]. A younger age, attendance to nursery or school, number of siblings, and unfavorable social conditions have all been described as predisposing conditions to NPC [1,3,4]. Many factors, such as antibiotic pressure, temporal trends, and the introduction of new conjugate vaccines, may explain the replacement of serotypes and changes in the rate of nasopharyngeal colonization [1,2,3].

Currently, 100 pneumococcal serotypes based on the capsular polysaccharide have been identified. These serotypes vary in terms of incidence, disease manifestation, and antibiotic resistance. Since the introduction of pneumococcal conjugate vaccines (PCVs), a global decrease in NPC has been observed for this pathogen, especially of vaccine serotypes (VSs) [4,5]. In Spain, the use of PCVs in children has been heterogeneous among the entire country. The 7-valent pneumococcal conjugate vaccine (PCV7) was commercialized in 2001, although mainly in the private market. The 10-valent pneumococcal conjugate vaccine (PCV10) was authorized in 2009 without generic use. The 13-valent pneumococcal conjugate vaccine (PCV13) was made available in 2010 for the private market. In 2015, the Ministry of Health approved the systematic use of PCV13 for the pediatric population and gave a deadline until the end of 2016 to include it at the national level (2+1 schedule).

In children <2 years of age, the incidence in Spain by PCV13 serotypes declined from 26.91 cases in the pre-PCV13 period to 3.89 cases in the last period (2018–2019), which demonstrated a reduction of 89% of IPD. The incidence by non-PCV13 serotypes increased from 9.41 cases in 2009 to 17.19 cases in 2018–2019. In children aged 2–5 years old, the incidence by PCV13 serotypes declined from 14.55 in 2009 to 1.93 in 2018–2019, which resulted in an 88% reduction in IPD. Other European countries who used a 2+1 schedule have observed serotype replacement by non-PCV13 serotypes [6,7,8,9]. Treating S. pneumoniae involves a high use of antibiotics, especially in children under 5 years, and many countries have an elevated incidence rate of resistance to more commonly used antibiotics [3].

We conducted this study five years after the PCV13 vaccine was licensed in Spain, with the aim to describe the potential influence of vaccination and other factors on nasopharyngeal colonization, replacement of serotypes, and antimicrobial susceptibilities of pneumococcal strains.

## 2. Materials and Methods

### 2.1. Study Population

This multicenter transversal study was conducted in the Region of Murcia, located in the Southeast of Spain, with a population of 16,200 children at one year of age and 17,996 at four years of age according to the National Statistical Institute [10]. In collaboration with the Pediatrics Association of Primary Care of the Murcia Region (APERMAP), this study combined the participation of 60 primary care pediatricians in the region in all health areas and microbiologists related to hospitals in the areas where the collected samples were analyzed. With voluntary participation, the ratio between the number of pediatricians in each area and the number and characteristics of the population attended was taken into consideration. The study was performed in healthy children who attended for examination in the Healthy Child Program.

In this study we recruited healthy children around 1 year old (10–14 months) and around 4 years old (3.5–4.5 years) who were vaccinated and non-vaccinated with PCV13 that attended the pediatric community centers for health care programs. Children with comorbidities or a fever at the time of sample collection were excluded from the study. The period of enrollment covered two seasons including June–August 2014 (Summer 2014) and January–March 2015 (Winter 2015).

Data records included geographical area, date of sample collection, age, sex, PCV13 vaccination status, breastfeeding, school or daycare center attendance, consumption of antibiotics during the previous months, passive smoker, and number of siblings. Children that met the inclusion criteria were invited to participate and written informed consent was explained to and obtained from the parents.

### 2.2. Microbiological Processing of the Samples

From each child included in the study, a nasopharyngeal sample using a sterile swab with Amies medium (Swabs®, DELTALAB, Barcelona, Spain) was obtained by nursing personnel trained in the technique in the different participating centers. The samples were sent to the reference hospital in each area within the next three hours and were immediately cultured on blood agar plates with an optoquine disk. Agar plates were incubated for 24 h at 37 °C with 5% CO_2_. α-hemolytic colonies suspected of pneumococcus were identified according to their morphology and sensitivity to optochine. Definitive identification was performed using biochemical tests, according to the protocols of each center. After identification and sub-culture, the strains were stored at –80 °C in a skim milk Difco medium and sent to the reference hospital for antimicrobial susceptibility testing and the Spanish Pneumococcal Reference Laboratory for serotyping.

#### Antimicrobial Susceptibility Testing

The agar dilution method was used to determine the minimal inhibitory concentrations (MIC) of nine antibiotics (penicillin, amoxicillin, cefotaxime, vancomycin, erythromycin, tetracycline, levofloxacin, chloramphenicol, and linezolid). Breakpoints for MIC and the disk diffusion method were those defined by The European Committee on Antimicrobial Susceptibility Testing (EUCAST, 2019) [11]. The disk diffusion method was used to test the activity of clindamycin, rifampicin, and cotrimoxazole, in accordance with the guidelines established by the Clinical and Laboratory Standards Institute (CLSI, 2018) [12].

Multidrug resistant strain (MDR) was defined as nonsusceptibility to three antibiotics, one of the three being a betalactam drug.

Serotyping was performed by Quellung reaction and dot blot assay [13] using specific antisera from the Statens Serum Institute (Copenhagen, Denmark) and/or by a molecular capsular sequence typing methodology [14].

### 2.3. Statistical Analysis

The sample size was calculated with a strength of 80%, for an expected proportion of carriage of 20%, with a 95% confidence interval and an accuracy of 2%, with the sample size needed being 1537. The selection of the sample was stratified according to the population attended by each of the seven hospitals with pediatric attention in Murcia.

The association between qualitative variables was evaluated by means of contingency tables analysis using Pearson’s Chi^2^ statistic and analysis of residuals, if required. Student’s *t* test was used for comparing means of the qualitative variables. Multivariate logistic regression analysis was carried out considering pneumococci isolation (yes/no) as the dependent variable and the following as independent variables: age (1 year/4 years), sex, number of doses of PCV13 vaccine, daycare attendance (yes/no), number of months of breastfeeding, number of siblings, and antibiotic use (yes/no) in the last month. Moreover, logistic regression was performed stratified by age (1 year and 4 years), and considered the same dependent and independent variables. Statistical significance for all results was set for a *p* value < 0.05.

## 3. Results

### 3.1. Pneumococcal Serotypes in Asymptomatic Carriers

A total of 1821 children were enrolled in the study, 893 boys and 928 girls (Table 1). Among them, 906 were children of one year (401 summer 2014, 505 winter 2015) and 915 of four years (411 summer 2014, 504 winter 2015). A large proportion of children (71.9%; 1309/1821) had received at least one dose of PCV13 and 67.8% (1235/1821) had received two or more doses. The total percentage of NPC was 17% (138/812) in summer and 21.9% (221/1009) in winter (*p* = 0.0089). The mean value for the whole group was 19.8% (359/1821) (Table 1). Nasopharyngeal colonization was similar in children aged one or four years (18.6%; 169/906) and (20.7%; 190/915), respectively (*p* = 0.278).

Serotyping could only be performed in 263 strains and the sensitivity study in 255. The serotypes’ distribution is shown in Figure 1, grouped according to their inclusion in the PCV13 or in the 15-valent pneumococcal conjugate vaccine (PCV15) and 20-valent pneumococcal conjugate vaccine (PCV20), which are currently in development. Only 38 (14.4%) of the serotypes found were PCV13-specific serotypes (VS). The persistence of 19F (13), 19A (7), 6A (6), 23F (4), 3 and 6B (3), and 9V and 14 (1) stands out. Serotypes 4, 18C, 1, 5, and 7F were not found. Eleven strains (4.3%) were non typeables and 206 (80.8%) were non-PVC13 serotypes (NVS), with special mention to the more than 20 isolates of serotypes 23B (34) and 11A (22), and more than 9 of 10A (17), 35F (15), 23A (14), 6C (13), 21 and 15A (10), and 16F, 24F, and 35B (9). In our population, the PCV15 and PCV20 serotypes accounted for 19.6% and 38% of those pneumococcus detected, respectively.

To analyze the NPC in general, univariate analysis was performed including variables such as number of siblings, PCV13 vaccination with at least one dose, and administration of antibiotics in the last month. Summer season was negatively associated with carriage (*p* = 0.04), and day care or school attendance was significantly associated with NPC in all groups (*p* = 0.006).

No significant statistical differences were found regarding sex, age, vaccination, recent antimicrobial use, passive smoking, or number of siblings; however, the one-year-olds vaccinated with any number of doses and those who had taken antibiotics presented a lower percentage of carriage.

A multivariate analysis by means of logistic regression was carried out, for all of the NPC isolates, including the following independent variables: age, sex, number of doses of vaccine PCV13, season, daycare attendance, number of months of breastfeeding, number of siblings, and antibiotic use in the last month. Summer season was found to be a protective factor (odds ratio (OR): 1.35. CI (1.05–1.73)), *p* = 0.019 and daycare or school attendance was a favoring factor for NPC (OR: 1.78. CI (1.24–1.53)), *p* ≤ 0.005 (Table 2).

### 3.2. Relationship between Antibiotic Resistance and Serotypes

Among the 255 carriage isolates, 35.7% (91 strains) had an MIC to penicillin > 0.06 µg/mL, 1.5% presented a high level of resistance (MIC > 2 µg/mL), and 34.1% were susceptible with increased exposure (sensitive to high doses of penicillin intravenous i.v.), in non-meningitis. The percentages of penicillin resistance, in accordance with the EUCAST criteria shown in Table 3.

Resistance to amoxicillin (MIC > 1) was 10.5% and 231 strains (90.6%) were sensitive to cefotaxime (MIC ≤ 0.5 µg/mL), 23 (9.1%) had intermediate sensitivity, and only one isolate (0.4%) was fully resistant (MIC > 2 µg/mL).

Erythromycin resistance accounted for 32.1% of isolates (82/255) and resistance to clindamycin was detected in 67 strains (26.3%), which indicates that the most frequent resistance mechanism was MLSB (Macrolide-Lincosamide_Streptogramine B). An M phenotype was present in 5.8% of the strains. Trimethoprim sulfamethoxazole (TMS) resistance was present in 28.2% (72 strains). Following the EUCAST 2019 criteria, no resistance to linezolid, levofloxacin, and vancomycin was found. Three isolates were resistant to rifampicin.

Table 4 show the highest resistant serotypes among the most frequent serotypes. The non-vaccine serotypes 23B (17/34), 11A (11/22), 35B (7/9), and 6C (8/13) and the 13 PVC-specific serotypes 19F (9/13), 19A (5/7), 6A (4/6), and 23F (3/4) were resistant to penicillin. The 65% of the VSs versus 31% of the NVSs were resistant to penicillin (*p* < 0.001).

Resistance rates to erythromycin were around 50% in the NVS 15A, 11A, 23A, 23B, and 35B. NSV 6C and VS 19A, 19F, and 6A exceeded 65% of resistant isolates.

Antimicrobial resistance to penicillin and erythromycin was found in 48 strains (18.25%), particularly in the VSs 19A, 19F, 6A, and 6B and NVSs 35B and 6C. Multidrug resistance was detected in 21 strains (8%) and significantly higher among PVC13 serotypes (19A, 6A, and 6B), and NVS 6C.

## 4. Discussion

It is important to know the prevalence of the pneumococcal serotypes colonizing infants of one year old, the population with the highest burden of IPD, and in those aged four years, the age with the highest incidence of nasopharyngeal colonization [1,15]. It is also interesting to know the potential role of vaccination in pneumococcal nasopharyngeal carriage.

NPC and vaccination: It is difficult to assess the impact of vaccination on NPC due to the lack of uniformity in populational studies. Although some studies have communicated a reduction in the percentage of NPC [4,5], most conclude that there are no important variations [16,17], given that after the drop in the VSs an increase in other serotypes has been observed, although not comparable to that observed with the serotype 19A after the introduction of the 7-valent pneumococcal conjugate vaccine (PCV7).

In our study, the global prevalence of pneumococcal colonization was 19.9%. This percentage is lower than those described in Spain in this age range, both prior to the introduction of PCV7 [18] as well as subsequently, with the vaccine already on the market, [19,20], all of which presented figures of around 30–35%. Other studies provide similar data, even in sequential time studies [16,21,22,23,24]. Some authors have reported higher percentages in specific populations [5,25,26,27]. Justifying these differences is not an easy task, but it is possible that there are multiple factors involved such as ethnicity, study period, daycare attendance, number of siblings, occupation of the pharyngeal niche by other bacteria, vaccination, time trends, etc. In our study, about half of the samples were collected in summer, a low proportion of infants under one year of age were attending daycare and had a very low number of siblings; these factors could contribute to having a lower NPC rate than in most studies.

According to all published studies [2,22,28], the percentage of VSs was low at 14.4%, but higher than those observed in other countries [4,5,25,26,27]. It is possible that these variations are related to the level of vaccination in the population, which in our study is not excessively high. Among the vaccine serotypes, the persistence of 19F, 6A, 6B, 3, and 19A should be highlighted and serotypes 4, 18C, 1, 5, and 7F were not found in accordance with other studies [2,29], except in the persistence of 19F.

The most frequently encountered NVSs were 23B, 11A, 10A, 35B/F, 23A, 6C, 21, 15A/B/C, and 31, some of which are referred to in the literature as having a high resistance rate to antibiotics [29]. Similar values have been described in other studies [2,17,22,25,27,29].

The importance of replacement by NVSs must be seen in time. In the first few years after the introduction of PCV13, a replacement by serotypes with decreased invasive potential was observed. This trend has been reversed and recent studies show an increase in invasive pneumococcal disease (IPD) in people over 15 years of age, particularly in those over 65 [30,31]. The serotypes primarily involved in this change are 8, 9N, and 12F, which are very invasive despite being rarely detected in carriers, whilst the most frequent in NPC belong to the least invasive group [31]. An increase in the IPD rates was also observed in France [6] with a similar pattern that affected, among other age groups, those under the age of two. In Spain, a reduction of 88% in IPD cases has recently been notified in the pediatric population for the period 2009–2019 [29]. In 2015, the most prevalent serotypes in IPD in Spain were 24F, 23B, and 10A [7], thus supporting the data obtained at a regional level where the serotypes 23B and 10A are among the four most frequent. In our study, the serotype 24F was not found to be among the most frequent, in contrast to the recent data for IPD in France and Spain, which show that the serotype 24F is the most frequent in the pediatric population [6,7]. One possible explanation for this is that the serotype 24F is a serotype that is a transitory colonizer of the pediatric nasopharynx for a short time, which conversely has a high capacity to produce IPD, similar to serotype 8, and was not observed in our study or related to the increase in invasive diseases in adults.

Considering our population, the potential coverage of PVC15 and PVC20 was 18% and 36.5%, respectively.

Monitoring the evolution of serotypes and their level of resistance to antibiotics is mandatory to the design of new strategies, particularly taking into account that neither serotypes 24F, nor 23B, are included in current vaccines or in those under development. 

NPC associated factor: A series of mainly environmental and socioeconomic factors associated with colonization have been classically recognized to be associated with NPC, such as age, daycare attendance, overcrowding, and passive smoking. On the other hand, vaccination, breastfeeding, and the administration of antibiotics in the previous month have been reported as protective factors [1,3,4]. In our study, we found, at a statistically significant level, that attending daycare or school was a predisposing factor to NPC and that summer season was a protective factor. Other authors found one or several of those factors [4,32], but situations involving overcrowding, mainly daycare attendance, coincide in all the studies. It seems logical to think that contact with a larger number of possible carriers increases the possibility of NPC.

NPC and antibiotic resistance: Antibiotic resistance patterns vary considerably from country to country and moreover evolve over time [1]. In Europe, Spain, together with France and Italy, presents one of the highest rates of resistance [33] with a general decline in the resistance to penicillin, cephalosporins, macrolides, and Trimethoprim sulfamethoxazole (TMS) following the introduction of the PCV. This situation has remained stable in recent years in both the strains of carriers and in those from invasive disease [34,35,36]. The factors that have been related to such changes are the use of antibiotics, introduction of new vaccines, reduction in pneumococcal diseases, and replacement by more resistant serotypes, in addition to other less-known factors such as the expansion of clones, etc. [1,17,34,36].

Using breakpoints for meningitis, 35.7% of the strains were resistant to penicillin (MIC > 0.06). High resistance to penicillin (non-meningitis breakpoints) was found in 1.6% (MIC > 2), and 34.1% of strains had intermediate resistance (MIC > 0.06 - <2). These data are similar to those found in numerous studies carried out in different scenarios [2,25,37,38,39,40,41,42]. Other studies have published higher rates (about 40%) [5,24] and the lowest rate has been reported in Switzerland (16.8%) [36]. In Spain, the European Centre for Disease Prevention and Control (ECDC) [35] reported 23.5% resistance in isolates from IPD in 2015. This percentage fell to 18.5% in 2018.

We have found a higher rate of resistance to penicillin among PVC13 serotypes (65%) against NVS (31%) than recorded in the literature [5], which reflects the impact of the pneumococcal immunization on the rates of penicillin resistance.

Resistance to oral amoxicillin was 10.5% (MIC > 1), a higher percentage than that described in France [25] and probably linked to the consumption of this antibiotic in our area.

A total of 90.5% of our strains were cefotaxime-susceptible with MIC ≤ 0.5 and only one isolate had an MIC > 2, percentages that concur with other communications [37]. Other authors [36,40] have found susceptible rates that exceeded 95% for meningeal infection, and in the other extreme Korea had a 72.3% susceptibility rate for meningeal infection [22].

Resistance to erythromycin is linked to that of penicillin and varies greatly [1]. In our study, we found a 33.1% resistance during the commercialization of PCV7, slightly lower than the rates published for other Spanish studies (40.5% and 37%) [20,43]. In other studies, a similar value can be found in Cyprus, among the Bedouins of Sokora, and in Russia [5,40,42] with higher resistance rates in France and the United States of America [2,38]. This is closely linked with the elevated consumption of macrolides [1,43,44].

The combined resistance to penicillin and erythromycin is a problem that appeared in the 1990s and that reached very high levels in Spain at the start of this century [45]. Recent years have seen a progressive drop in this type of isolate. The data from our study are somewhat higher (18.2%) than those informed by the European Center for Disease Prevention and Control (ECDC) [35] for Spain, with a rate around 12%, which has not changed significantly in the last few years.

Resistance to TMS reached 27.7%, similar to Cyprus [40] and the United States of America [38], and lower than the values found by other authors (35–45%) [27,41,42].

The rate of multiresistance was similar to that described in Spain (6%) and much lower than the data from France [25].

### Strengths and Weaknesses

The main strengths of this work lie in the moment in which it was performed, five years after the commercialization of PCV13, and the number of samples gathered, despite the excessive number of samples lost. Another strong point is that the epidemiological data were gathered from clinical records, thus not having to rely on parents’ memory, except on a very few occasions, and thereby avoiding bias in the face of “sensitive” questions such as smoking or the duration of breastfeeding. However, as often occurs in this type of study, there was no prior randomization, given that participation was voluntary, so it may not be a totally representative sample. Data regarding the doses and duration of prior antibiotic therapy were not collected, nor was multiple colonization assessed, which may have modified the final results.

## 5. Conclusions

We have found a relatively low percentage of VSs, which confirms the importance of pediatric vaccination for reducing the carrier state by vaccine serotypes, with a percentage of NPC that is also low with respect to other studies. We observed the permanence of VSs, principally 19F, the non-appearance of 4, 18C, 1, 5, and 7F, and an increase in the NVSs, principally 23B, 11A, 10A, 35B/F, 23A, 6C, and 21. Attendance to daycare or school is a NPC-favoring factor. The remaining VSs have a high degree of resistance and among the NVSs, special mention should be made to 35B, 11A, 23B, and 6C. A high rate of resistance to penicillin, erythromycin, and trimethoprim sulfamethoxazole was found.

It is necessary to monitor the evolution of the serotypes and the resistance of S. pneumoniae to commonly used antibiotics to enable the design of new vaccines and the implementation of public health measures.

## Figures and Tables

**Figure 1 vaccines-09-00014-f001:**
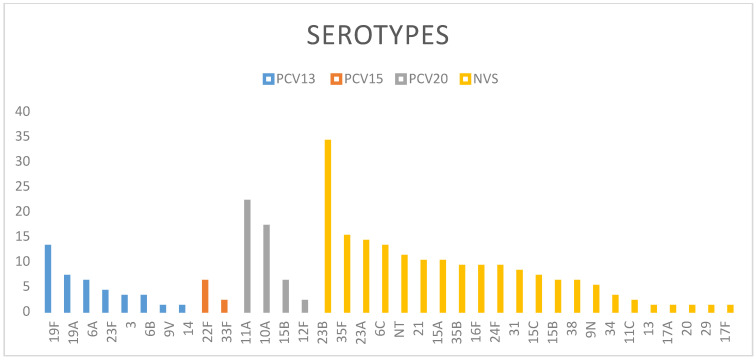
Serotype distribution of nasopharyngeal Streptococcus pneumoniae isolates from healthy children in Murcia. PCV13: pneumococcal conjugate vaccine 13-valent. PCV15: pneumococcal conjugate vaccine 15-valent. PCV20: pneumococcal conjugate vaccine 20-valent. NVS: non-vaccine serotypes. NT: non typeable serotypes.

**Table 1 vaccines-09-00014-t001:** Social and demographic characteristics of children.

Characteristic/Variables	Frequency N (%)		
	2014	2015	Total
No. of children	812 (445)	1009 (55.4)	1821
Age			
1 year	401 (49.4)	505 (50)	906 (49.7)
4 years	411 (50.6)	504 (50)	915 (50.3)
Gender			
Male	350 (43.1)	543 (53.8)	893 (49)
Female	462 (56.9)	466 (46.2)	928 (51)
Nasopharyngeal carriage	138 (17.1)	221 (21.9)	359 (19.8)
Siblings ≥ 2	122 (15)	131 (13)	253 (13.9)
School attendance	484 (59.6)	599 (59.4)	1079 (59.2)
Day Care attendance 1 year	73 (9)	95 (9.4)	168 (9.2)
Antibiotics during last month	55	114	169 (10.4)
Aged 1 year	26	56	
Aged 4 years	29	58	
Breastfeeding ≥6 months	324 (39.9)	333 (33)	657 (36)
Smoke exposition	323 (39.7)	393 (38.9)	716 (39.3)
PCV13-Vaccinated ≥2 doses	585 (72)	650 (64.4)	1235 (67.8)

**Table 2 vaccines-09-00014-t002:** Multivariate analysis.

	Colonized	Not Colonized	*p*	Multivariate OR (CI95%)
Age (%):				
1 years	18.9	81.1		
4 years	21	79	0.28	0.73 (0.52; 1.03)
Sex:				
Male (%)	19.7	80.3		
Female (%)	20.2	79.8	0.78	0.98 (0.78; 1.25)
Vaccination (%):				
Full (Yes)	18.5	81.5		
Full (No)	21.6	78.4	0.26	0.77 (0.58; 1.02)
At least 1 dose (Yes)	18.5	81.5		
At least 1 dose (No)	20.7	79.3	0.27	0.85 (0.66; 1.11)
Day-care/School attendance (%):				
Yes	22.1	77.9		
No	16.7	83.3	0.005	1.78 (1.24; 1.53)
Breast feeding:				
number of months (Mean (SEM))	5.7 (0.4)	5.95 (0.7)	0.9	1.0 (0.99; 1.00)
Siblings (%):				
No	41.5	58.5		
Yes	47.2	52.8	0.055	1.16 (0.91; 1.47)
Antibiotics last month (%):				
Yes	20.3	15.6		
No	79.7	84.4	0.15	0.68 (0.43; 1.07)
Season (%):				
Summer	17.1	82.9		
Winter	21.6	78.4	0.019	1.35 (1.05; 1.73)

**Table 3 vaccines-09-00014-t003:** Antimicrobial susceptibility of 255 S. pneumoniae isolates to 12 antimicrobial agents (breakpoints from The European Committee on Antimicrobial Susceptibility Testing (EUCAST, 2019)).

Antibiotic	Susceptible Isolates n/%	Resistant Isolates n/%	Intermediate Isolates n/%
Penicillin (non-meningitis)	164/64.3	4/1.6	87/34.1
Penicillin (meningitis)	164/64.3	91/35.7	
Oral Amoxicillin	228/89.5	27/10.5	
Cefotaxime	231/90.6	1/0.4	23/9.1
Vancomycin	255/100		
Levofloxacin	255/100		
Erythromycin	173/67.8	82/32.1	
Tetracycline	187/73.3	68/26.7	
Chloramphenicol	244/95.7	11/4.3	
Clindamycin	188/73.7	67/26.3	
Linezolid	255/100		
Trimetroprim-sulfamethoxazole	183/71.8	72/28.2	
Rifampicin	252/98.8	3/1.2	

**Table 4 vaccines-09-00014-t004:** Resistance rates to penicillin, erythromycin, and trimethoprim sulfamethoxazole among the most frequent serotypes.

Serotype	Number of Isolates	Penicillin Intermediate and Resistant Isolates n/(%)	Erythromycin Resistant Isolates n/(%)	Trimethoprim Sulfamethoxazole Resistant Isolates n/(%)
11A	22	11/(50)	10/(45.5)	16/(72.7%)
23B	34	17/(50)	7/(43.8)	14/(41.2)
19F *	13	9/(69)	9/(69)	9/(69.2)
6C	13	8/(61.5)	8/(61.5)	3/(23)
35B	9	7/(77.7)	4/(44.4)	1/(11)
19A *	7	5/(71.5)	5/(71.5)	2/(28.5)
6A *	6	4/(66.6)	4/(66.6)	5/(83)
15A	10	2/(20)	5/(50)	0/(0)
23A	14	1/(7)	7/(50)	4(28.5)

* Vaccine serotypes.

## Data Availability

The data presented in this study are available in insert article.

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
