# Peer review of "Impact of Pneumococcal Vaccination in the Nasopharyngeal Carriage of Streptococcus pneumoniae in Healthy Children of the Murcia Region in Spain"

_vaccines, 2020, doi:10.3390/vaccines9010014_

Round 1

Reviewer 1 Report

Dear Editor and Authors:

I have carefully read this manuscript. et al studied the prevalence of VS and its relationship to Streptococcus pneumoniae vaccination in a group of kids in Spain. The antibiotic susceptibility of circulating Streptococcus pneumoniae strains was also studied which gave suggestions to future vaccine design.

Overall, this study is meaningful. However, this study needs some serious revision in order to be publishable.

Major concern:

The language needs to be improved. This paper is difficult to follow. I suggest to add subtitles, organize related contents into one paragraph and summarize the paragraph in its first sentence.

ie countrys , datas are not proper English expression.

Author Response

Dear reviewer:

Thanks for your review : Thank you for your review. You will find the answers to your requests below. 

  1. The language needs to be improved. This paper is difficult to follow. I suggest to add subtitles, organize related contents into one paragraph and summarize the paragraph in its first sentence.

First of all, we have done an extensive revision of english lenguaje and then we have added subtitles in the discussion, in order to maKe reading easier

  1. Does the introduction provide sufficient background and include all relevant references?

We have added in the introduction some background and references that we think may be relevant.

Thank you

Reviewer 2 Report

This regional study on evolution of the pneumococcal serotypes and their level of resistance over a large sample is of interest and is likely to be able to inform the vaccine strategies to be implemented. Antibiotic resistance data are also of interest, even if data on previous antibiotic therapy is lacking. The manuscript is well written and is suitable for publication

Minor comments:

Add in the introduction a sentence on the incidence rate in Spain before and since the vac

Line 48 : Please provide a quantified estimate of children asymptomatic nasopharyngeal carriage rate which can be compared to the prevalence observed in the study

It would be interesting to describe the factors associated with the absence of PCV13 vaccination.

Table 1. N° Children: please check the numbers

The moderate reduction in carriage in vaccinated children merits further discussion

Line 167 : correct frecuent

Line 254 and 256 : please specify what percentage it is

Author Response

Dear reviewer:

Thanks for your review. You will find the answers to your requests below. 

Add in the introduction a sentence on the incidence rate in Spain before and since the vac

We have included data of pneumococcal disease incidence in Spain and the data of carriage of Streptococcus pneumoniae are included in the discussion

Line 48 : Please provide a quantified estimate of children asymptomatic nasopharyngeal carriage rate which can be compared to the prevalence observed in the study

We have included an approximate carriage rate

It would be interesting to describe the factors associated with the absence of PCV13 vaccination.

This data was not included as a variable in the study but we have performed a logistic regression analysis with the collected variable and we have found variability between the different participating health areas. This might be due to differences in the pediatricians advice respect the vaccination and different socieconomic and cultural levels, data not collected. It is not possible to deduce it in this study.

Table 1. N° Children: please check the numbers

We have reviewed and corrected some errors in the tables

The moderate reduction in carriage in vaccinated children merits further discussion

 It is difficult to explain the low rates of pneumococcal carriage found in our study respect  other reports. In the discussion, we commented the factors that, in our opinion, have been able to have a decisive influence in this data such as the low schooling of 1-year-olds childrens and the high proportion of samples collected during summer.

Line 167 : correct frecuent

corrected 

Line 254 and 256 : please specify what percentage it is

The % has been set

Thank you

Reviewer 3 Report

Introduction is very brief and importance of vaccination, reason for the antimicrobial susceptibility studies can be elaborated. 

why the dose and duration of antibiotic therapy was not collected ?

what is  the novelty in the present study ? 

Author Response

Dear autor:

Thanks for your review. You will find the answers to your requests below. 

English language and style are fine/minor spell check required

We have proofread the manuscript again and spelling errors have been corrected.

Does the introduction provide sufficient background and include all relevant references?

As suggested by the reviewer the Introduction section has been broadened and new references have been added. We have included additional information about the impact of pneumococcal vaccination in the prevention of invasive pneumococcal disease. We have also address the role of pneumococcal colonization and we have included some text related to the potential role of this pathogen on the antimicrobial resistance.

Is the research design appropriate?

We consider that the study design allows us to address the research question. Despite the that the study is not totally randomized,  the ratio between the number of Pediatricians in each Area and the number and characteristics of the population attended was taken into consideration.

Are the methods adequately described?

In order to better described the methods used, we have added a paragraph explaining the data collection. Additional information on how samples were collected from the health clinics has also been added.

Are the conclusions supported by the results?

We consider that the study results support the conclusions.

We consider that our conclusions are based solely on the study results. Nevestheless, we could explain further if you specify what is unclear.

Introduction is very brief and importance of vaccination, reason for the antimicrobial susceptibility studies can be elaborated. 

As already stated in point 2 the Introduction section has been broadened in order to highlight the benefits asociated with pneumococcal vaccination and the problem of antibiotic resistance

why the dose and duration of antibiotic therapy was not collected ?

There are twoo main reasons why we have not considered important to collect information about dose and duration of antibiotic. First of all, previous studies suggest that the consumption of antibiotics itself is more important than dose and duration, so we did not collect that information. In addition, since our study included only healthy children, the number of participans with antecedent antibiotics use were very low.

what is  the novelty in the present study ? 

Even though the present study is not enterly novel we are convinced of the importance of our results. Monitoring the distribution of serotypes causing pneumococcal disease among children is crucial in the development of future conjugate vaccines. Furthermore, to analyze the changes in the epidemiology of pneumococcal disease is required to evaluate the effectiveness of the current vaccination strategy.

Thank you

Round 2

Reviewer 1 Report

Dear Authors:

Thanks for revision. I do not have further objections. As a result, I suggestion acceptance of this version.